## Research Article

bipolar; carers; developing countries; families and mental health; global mental health

**Corresponding author:**
Madeha Umer;
Email: madeha.umer@mail.utoronto.ca

# Lived experiences of bipolar disorder and family caregiving in Pakistan

Madeha Umer[1] , Muqaddas Asif[2], Ameer Bukhsh Khoso[2], Brett D.M. Jones[3,4], Iqra Hassan[2], Siqi Xue[3] , Sonia Langhani[2], Cindy-Lee Dennis[5,6], Farooq Naeem[3,4], Benoit Mulsant[4], Nasim Chaudhry[2], Nusrat Husain[7] and Muhammad Ishrat Husain[4]

[1]Institute of Medical Sciences, University of Toronto, Toronto, ON, Canada; [2]Pakistan Institute of Living and Learning, Karachi, Pakistan; [3]Campbell Family Mental Health Research Institute, Centre for Addiction and Mental Health, Canada; [4]Department of Psychiatry, Temerty Faculty of Medicine, University of Toronto, Toronto, ON, Canada; [5]Lawrence S. Bloomberg Faculty of Nursing, University of Toronto, Toronto, ON, Canada; [6]Lunenfeld-Tanenbaum Research Institute, Toronto, ON, Canada and [7]University of Manchester – The Victoria University of Manchester Campus, UK

## Abstract

Bipolar disorder (BD) is associated with substantial disability and caregiver burden. In Pakistan, prevalence is unusually high, and limited mental health services place families at the center of care. Cultural and religious beliefs strongly shape how the illness is recognized, understood and managed. This study explored how individuals living with BD and their caregivers understand and navigate the illness within this context. Semi-structured interviews were conducted with 12 adults diagnosed with BD (type I or II) and 12 caregivers recruited through a national registry. Open-ended questions explored illness understanding, caregiving challenges, cultural influences and preferences for family intervention. Interviews were conducted in Roman Urdu, audio-recorded, transcribed verbatim and analyzed using Braun and Clarke's reflexive thematic analysis. Purposive and snowball sampling ensured diversity in gender, socioeconomic status and residence. Patients described blended biomedical and cultural explanations of BD, symptom-related disruption, treatment barriers, stigma and coping through routines and religious practices. Caregivers reported confusion at illness onset, financial and emotional burden, inconsistent support and the need to navigate biomedical and spiritual care pathways. Both groups emphasized the need for accessible, family-inclusive interventions. The findings support development of culturally tailored, scalable and faith-sensitive family interventions.

## Impact statement

Bipolar disorder is often managed within specialist mental health systems, yet in Pakistan, the reality is different: families provide most day-to-day care while services are limited, costly and difficult to access. This study brings together the voices of people living with bipolar disorder and their caregivers to show what support looks like in a low-resource, collectivist setting and why many existing care models do not translate well without adaptation. Our findings highlight that illness understanding and help-seeking are shaped by intertwined biomedical, cultural and spiritual explanations. Families commonly use plural pathways to care, combining psychiatric treatment with faith-based or spiritual practices. These approaches are not simply alternatives to medical care; they often function as sources of meaning, hope and coping, especially when formal services feel inaccessible or inadequate. Caregiving emerged as long-term and gendered, affecting finances, health, relationships and social standing. Stigma and secrecy can reduce community support and delay effective treatment, while practical barriers (travel, costs, fragmented services) intensify strain. At the same time, families described strategies that help them endure: routine-building, relational support and religious coping, suggesting clear entry points for intervention. By mapping these real-world experiences dyadically, this study provides actionable direction for designing scalable, culturally tailored and faith-sensitive family interventions for bipolar disorder. The results can inform service planning, clinician training and community-facing psychoeducation that respects local belief systems while improving access, continuity and caregiver support in Pakistan and similar contexts.

## Introduction

Bipolar disorder (BD) is a chronic psychiatric condition marked by recurrent mood episodes, functional impairment and high disability burden worldwide (Bipolar Disorder, 2025). Global lifetime prevalence estimates range between 0.4 and 1.1% (Merikangas et al., 2011), yet recent evidence suggests higher rates in South Asia (Vidyasagaran et al., 2023), with Pakistan's first

nationally representative survey reporting a prevalence of 4.5% (Rahman et al., 2024). This figure indicates that nearly one in 20 adults in Pakistan is living with BD, underscoring the condition's public health significance.

Scarce mental health resources and systemic underinvestment magnify the burden of BD in Pakistan. Pakistan allocates only 0.4% of its health budget to mental health, with a psychiatrist-to-population ratio of 0.19 per 100,000, among the lowest in South Asia (Main Thompson and Saleem, 2025). The economic costs are equally striking: the annual burden of mental illness was estimated at PKR 616.9 billion (~USD 3 billion) in 2020, borne mainly by families through out-of-pocket expenditure and lost productivity (Alvi et al., 2024).

Given the limited availability of mental health services, families often serve as the primary caregivers for individuals with BD in Pakistan. Caregiving encompasses practical supervision and financial and emotional support, and is usually accompanied by significant personal sacrifice and societal stigma (Siddiqui and Khalid, 2019). This is particularly important as existing research highlights the bidirectional relationship between patient BD outcomes and caregiver well-being. For example, symptom recurrence exacerbates caregiver stress, while caregiver burden contributes to relapse risk (Miklowitz, 2007; Van Der Voort et al., 2007). Most of this evidence, however, comes from high-income settings with greater access to mental health services. These dynamics remain underexplored in low-resource, collectivist settings such as Pakistan, where cultural and religious frameworks profoundly shape illness perceptions and care pathways (Shah et al., 2019; Shafiq, 2020; Umer et al., 2025).

Despite the centrality of families in sustaining care, qualitative research on how patients and caregivers experience and navigate BD in Pakistan is limited. Understanding these perspectives is critical for designing culturally relevant and feasible interventions. This study aimed to explore the lived experiences of individuals with BD and their family caregivers in Pakistan, focusing on how illness is recognized, managed and interpreted within cultural, social and systemic constraints.

## Methods

### Study design

This qualitative study engaged individuals with BD and their caregivers across Pakistan. This study was approved by the National Bioethics Committee of Pakistan (Ref. No. 4–87/NBC-970/23/457). Semi-structured interviews were conducted using interview guides (Supplementary Materials) developed from established qualitative methodologies (Naeem et al., 2012; Naeem et al., 2016). These interviews were conducted separately with patients and caregivers to explore their lived experiences and perspectives related to BD and family-based care. The guides were piloted with three patient–caregiver dyads to identify emergent themes and refined iteratively to capture culturally relevant insights. The patient interviews explored participants' understanding of BD and its causes, challenges in daily life and relationships, experiences of stigma and treatment, perceived family support and expectations from a family intervention. The caregiver interviews focused on caregiving experiences, the impact of BD on family life, cultural and religious influences on caregiving, access to professional and informal support, and views on the design and feasibility of a family intervention program. Across both groups, open-ended questions encouraged participants to share personal narratives and reflections on family dynamics, coping strategies and unmet needs. The study aimed to generate an in-depth, contextually grounded understanding of shared and cross-cutting experiences of living with BD and family caregiving in Pakistan, rather than to compare experiences across participant subgroups. Data collection and analysis followed a concurrent approach, allowing ongoing refinement of questions. The research team was trained in Good Clinical Practice (ICH-GCP) and ethical interviewing techniques.

### Recruitment

Participants were recruited nationwide between May 2024 and April 2025 *via* the National Bipolar Disorder Registry maintained by the Pakistan Institute of Living and Learning (PILL), a research nonprofit organization in Pakistan. Purposive sampling ensured diversity in gender, socioeconomic status and urban/rural residency. Snowball sampling supplemented recruitment to achieve saturation (Naderifar et al., 2017). Sample size was guided by qualitative principles of information power/thematic sufficiency (Malterud et al., 2016). Recruitment and interviewing ceased when additional interviews were not generating substantively new insights relevant to the study aims.

### Participants

The patient group consisted of twelve adults aged 18 years or older who had been clinically diagnosed with either bipolar I disorder (BD-I) or bipolar II disorder (BD-II) by consultant psychiatrists. All participants were confirmed to be in a remission phase, as indicated by scores of 12 or below on the Young Mania Rating Scale (YMRS) and 10 or below on the Montgomery–Åsberg Depression Rating Scale (MADRS). Each participant was also assessed to be capable of providing informed consent prior to enrollment in the study. The caregiver group included 12 family caregivers, each aged 18 years or older, who had been providing unpaid care to an individual with BD for at least one year. Caregivers who were simultaneously responsible for individuals with other severe medical or psychiatric conditions were excluded from participation.

### Study procedure

Potential patients and their caregivers received study information from a trained research assistant (RA) in person or by phone. All participants were given the option to read and discuss their queries with family or the research team before deciding whether to participate.

After obtaining consent, the RA scheduled the interview date and time based on the participant's availability. Each participant completed one semi-structured interview lasting approximately 60–90 min, conducted in person or by telephone depending on participant preference and facilitated by the lead researcher. Immediately following each interview, the lead researcher recorded brief field notes and reflexive memos, which were used alongside the transcripts during familiarization and subsequent theme development. Each participant received PKR 500 as compensation for their time immediately following the interview. All interviews were audio-recorded, transcribed verbatim (Roman Urdu script) and anonymized. Transcripts were stored on encrypted PILL servers with restricted access. Trustworthiness was supported using strategies aligned with qualitative criteria of credibility, dependability, confirmability and transferability (Nowell et al., 2017). An audit trail was maintained to document recruitment procedures, interview guide refinements, analytic decisions and theme development.

Candidate themes were iteratively checked against the coded extracts and the full dataset to ensure coherence and to avoid overgeneralization. To support transferability, we provide detailed participant characteristics (Supplementary Table 3) and include illustrative quotations in the Results to demonstrate how interpretations are grounded in participants' accounts.

### Data analysis

The analysis employed Braun and Clarke's (2006) six-phase reflexive thematic analysis framework, strategically incorporating artificial intelligence (AI) assistance while maintaining rigorous human oversight for cultural interpretation. During phase 1 (familiarization), the lead researcher (a native Urdu speaker) immersed themselves in the data by repeatedly reading the transcripts, relistening to audio recordings and reviewing accompanying field notes and reflexive memos. Because the same lead researcher conducted the interviews and led the analysis, contextual observations documented during and immediately after interviews informed interpretation during coding and theme refinement. Consistent with the study aims, coding and theme development prioritized shared meanings across participants rather than subgroup comparisons. This deep immersion established essential cultural grounding before introducing computational tools.

Phase 2 (coding) involved generating initial semantic codes across the dataset. To assist with this step, anonymized Roman Urdu transcripts were processed in ChatGPT-4.5 (ChatGPT, 2025) using standardized prompts to produce draft, surface-level codes; these outputs were then reviewed, edited and verified by the lead researcher for accuracy, cultural nuance and contextual fit. AI outputs were treated as provisional suggestions rather than analytic decisions. This coding workflow is described in detail in a separate methodological paper. Phase 3 (searching for themes) involved organizing the verified codes into candidate themes and subthemes, using manual clustering and semantic mapping to identify coherent patterns (*e.g.*, grouping codes relating to shame, concealment and social consequences under a candidate theme such as 'Family Stigma Dynamics'). Phase 4 (reviewing themes) involved iterative testing of preliminary themes against the whole dataset, refining constructs such as 'Spiritual Coping' after identifying urban–rural variations in practices. Phase 5 (defining and naming themes) involved articulating the central organizing concept of each theme, specifying its scope and naming themes to reflect their cultural meaning.

Finally, phase 6 (producing the report) involved selecting illustrative extracts and developing an analytic narrative that situates themes within Pakistani collectivist and family systems contexts. All analytical processes, including transcript review, coding and thematic development, were conducted in Roman Urdu to ensure cultural and linguistic fidelity. For publication, illustrative quotes were professionally translated into English, retaining only those Urdu terms without direct English equivalents with contextual explanations in brackets. Throughout all phases, AI served strictly as a phase 2 coding assistant, with all interpretive acts, including theme construction, refinement and cultural sensemaking, remaining exclusively researcher-led to ensure contextual fidelity.

### Results

#### Participant characteristics

A total of 54 participants were approached, 30 consented, and 24 were interviewed, including 12 patients with BD and 12 caregivers. Patients ranged in age from 20 to 61 years, while caregivers ranged from 20 to 60 years. Most participants were married (n = 16), with a smaller proportion unmarried or divorced. Educational levels ranged from no formal schooling to university degrees (BS/BA), and employment ranged from informal labor and household roles to skilled professions (*e.g.*, businessman, teacher, salesman, cable worker). Reported monthly incomes reflected substantial socioeconomic diversity, from under PKR 2,000 (USD 6.9) to over PKR 65,000 (USD 224).

The duration of BD diagnosis among patients spanned from 2 to 25 years. Treatment histories indicated frequent reliance on multiple providers, with many reporting consultations with between two and nine doctors. Caregivers often had caregiving roles spanning 2 to 25 years, underscoring the long-term and intergenerational nature of family involvement in care. Nearly all participants identified as Muslim (reflecting national averages), and cultural identity was tied to linguistic diversity, including Urdu, Punjabi and Pashto. Residence spanned both urban centers (Karachi, Lahore, Peshawar) and rural/village contexts, with reported travel time to mental health services ranging from 30 min to over 2.5 h. Family sizes varied substantially, with many participants living in extended households of 6–15 members. The caregiver sample comprised primarily brothers (n = 4), followed by mothers (n = 3). Details of participant sociodemographic and clinical characteristics are summarized in Supplementary Table 3.

#### Individuals with bipolar disorder

Thematic analysis of interviews with individuals living with BD in Pakistan revealed five overarching themes that capture the complexity of their lived experiences. These themes reflect both universal challenges of BD and culturally specific nuances of navigating the illness in Pakistan. Within these themes, participants articulated distinct subthemes, explored in depth below, that highlight critical dimensions of their journeys. Illustrative quotes for each theme and subtheme are provided in Supplementary Table 1, in both Roman Urdu and English translation.

#### Theme 1: Understanding and explanatory models

##### Knowledge gaps and diagnostic desire
Early encounters with BD were often marked by confusion and uncertainty. Many participants described a lack of clear explanation about what the illness meant, leaving them unsure whether their shifting moods and energy were ordinary fluctuations or symptoms of something more serious. Formal diagnoses were rarely communicated in an accessible way; instead, people often assumed a diagnosis existed only when medication was prescribed. Even when a diagnosis had been made, some participants felt it was never clearly conveyed to them. *'No, maybe it's written in my documents, but I still do not know'*, one patient said, underscoring how diagnostic information could remain opaque. This lack of open dialogue created a sense of disconnection, leaving individuals to struggle to place their experiences within a recognizable framework of illness.

##### Biological, cultural and spiritual attributions
Participants offered many different explanations for the causes of their illness, ranging from the physical to cultural and spiritual. Some linked the onset of BD to past injuries or severe bodily trauma, such as accidents in childhood or head injuries, which they believed left lasting weakness in the brain. One participant described how an electrical injury in late adolescence shaped his understanding of mental weakness: *'Around age 18, I received*

an 11,000-volt shock, which initially weakened my mind a lot'. Others framed their understanding in terms of spiritual or cultural models, with ideas such as the evil eye or supernatural interference providing a more familiar explanation. As one patient emphasized, *'I would not call it a disease, "nazar" (evil eye) is something that can kill someone while alive, if it strikes, it squeezes the life out of you'*.

For some, substance use and lifestyle factors also played a role, shaping the sense that BD was both an individual and social problem. One participant attributed the onset primarily to substance use and rejected spiritual explanations: *'For me the issue was I was using drugs at first, people went to exorcists saying "saya" (paranormal procession) happened, but I do not believe that'*. Others integrated multiple perspectives, combining biological explanations with family history, personality traits and ongoing stress. Reflecting this layered meaning-making, one participant noted, *'In my opinion, all these things are mixed (as causes)'*, while another highlighted the role of stress sensitivity: *'if there's tension, it gets triggered more easily, and stress increases'*. Many emphasized the role of everyday pressures, noting that financial struggles, academic failure and domestic tensions could intensify symptoms or bring about episodes.

Life events carried deep significance in shaping these narratives. Divorce, harassment and loss were described not only as personal traumas but also as social turning points that intensified illness. The death of a child or the collapse of a marriage, for instance, were remembered as events that pushed participants into despair or triggered their first episodes. Another participant recalled that following a critical illness and bereavement, *'gradually, I went into depression'*, illustrating how major losses were woven into participants' explanations of how their illness began or escalated.

### Theme 2: Symptom experience and impact

#### Symptom phenomenology

Participants described mood episodes as overwhelming and challenging to manage. Disturbed sleep was among the most common experiences, with many reporting long periods of wakefulness followed by brief or fragile rest. As one patient explained, *'I stay awake all day and do not sleep at night, then in the afternoon I'll wake up by around 12'*, capturing a disruptive cycle of sleepless nights and daytime crashes. Even when medication helped temporarily, symptoms often returned quickly, leaving them caught in cycles of exhaustion and relapse. One participant expressed frustration with limited symptom relief despite taking medication: *'if I take a pill at night, by day I feel just as if I were mad, wandering around like a lunatic'*.

Cognitive changes were also prominent. Participants spoke of intrusive and racing thoughts that felt uncontrollable, along with memory lapses that disrupted daily functioning. Some described their minds as overactive, filled with ideas that did not translate into purposeful action, leading to frustration and conflict within the home. Emotional instability was another defining feature. Rapid shifts between agitation, irritability, sadness and hopelessness left participants feeling unpredictable and distressed. One participant described the sense of losing control: *'it feels like 90% of the time, my mind is not under my control, the anger really takes over'*. These mood changes often carried thoughts of self-harm or suicide. Physical sensations accompanied these emotional states, including headaches, burning sensations and auditory disturbances, such as *'hearing the sound of bells in my ears'*, which further blurred the line between mental and bodily suffering.

### Behavioral changes and everyday life disruptions

Participants described how BD disrupted many aspects of daily life, particularly work, education and family relationships. Mood instability often led to irritability and anger, which could escalate into conflict with others or self-harming behaviors. These episodes left individuals feeling guilty or remorseful but unable to prevent recurrence. One participant described becoming aggressive toward family members and then feeling regret afterward: *'sometimes I hit my children. Although I feel regret too, and I apologize to them afterward'*. The impact extended to school and work, where difficulties with concentration and emotional regulation sometimes resulted in disciplinary action or job instability. A younger participant described educational consequences: *'because of anger issues I could not study well, I was suspended from class for a week'*.

For others, mood episodes triggered periods of excessive activity and impulsive generosity, such as overspending or giving away money, which later led to financial strain and tension within families. One participant described oscillating between expansive spending and conflict when resources ran out: *'when I have money, I spend a lot, but when I run out of money, I get frustrated, fight with people'*. Work routines and self-care were often inconsistent, with bursts of productivity followed by inactivity or chaotic behavior. At times, participants also expressed a desire to escape entirely, including thoughts of running away or suicide. One participant described repeated urges to flee and self-harm: *'I would try to run away, I would say I want to jump into the river'*.

### Theme 3: Treatment needs and preferences

#### Navigating treatment access and challenges

Participants highlighted how navigating mental health care was shaped by persistent structural barriers. Long travel distances, particularly for rural residents, made routine appointments difficult and often unsustainable. Even when services were available, high consultation fees and the cost of medicines made treatment unaffordable, forcing some to rely on cheaper but less reliable alternatives. One participant noted, *'I have seen in private hospitals, but their fee is very high, they prescribe very expensive medicines, and I cannot afford it'*. Public hospitals, though more affordable, were often overcrowded, with rotating junior doctors and limited continuity of care, leaving families feeling dismissed and unsupported.

Despite the strain, some participants described resilience through informal coping strategies, such as adapting sessions to more convenient locations or maintaining personal connections with trusted therapists long after formal treatment ended. One participant emphasized the value of continuity and flexibility, noting, *'Even though they are not in regular practice now they still do sessions with me'*, and *'if needed we can meet anywhere like at a restaurant, and have a session there'*.

#### Adherence behaviors and relapse prevention

Participants described medication use as a balancing act between the benefits of symptom relief and the fears attached to dependency, stigma or adverse effects. Many acknowledged that medicines were essential for regaining stability, yet hesitated to adjust doses or continue long-term, reflecting a mistrust of both side effects and the idea of becoming reliant on drugs. As one participant explained, *'I did not increase that dose (when the doctor said so) because I do not want to become dependent'*. Decisions about adherence were often shaped by hierarchy and trust in providers; while prescriptions from senior clinicians were generally followed, advice from junior staff was more likely to be questioned or ignored. One participant

described this explicitly: *'I followed the medication prescribed by the surgeon, but I did not follow the higher dose suggested by the junior doctors'*.

Systemic and cultural contexts added further complexity. In some households, psychotropic drugs were taken casually or dismissed altogether, reflecting broader family attitudes. Others expressed religious or spiritual ambivalence, questioning whether long-term use was harmful or even necessary. Practical issues also undermined adherence, with public hospitals frequently rotating doctors and altering prescriptions, sometimes leading to destabilizing side effects. As one participant noted, *'when the doctor changes, the medicines change, some medicines help me feel normal while others make me sleepy and depressed'*.

Despite these barriers, many participants found ways to sustain treatment. Some relied on simple routines and reminders to maintain regular use, while others placed complete trust in their psychiatrist's guidance. The consequences of relapse, ranging from hospital readmissions to social and familial disruption, were substantial motivators to continue.

### Care modality and format preferences

Participants stressed the need for care that combined medical and psychosocial support. Medication was widely viewed as the foundation for stability, with talk therapy and other psychosocial interventions considered valuable but secondary. As one participant put it plainly, *'Talk therapy alone will not work properly until someone takes medicines prescribed by a general doctor'*, reflecting a strong belief that pharmacotherapy was a prerequisite for improvement. Many believed that professional guidance alongside medicines offered the most effective path to recovery. Group-based interventions were seen as especially useful for raising awareness among families unfamiliar with mental illness, providing opportunities to share experiences and feel less isolated. One participant noted: *'group interventions are very good especially for families who do not accept or know about mental health or who think it's all nonsense'*. At the same time, individual sessions were valued for the focused guidance they offered. A recurring theme across both formats was the relief of being able to express emotions openly, which participants described as making their burdens feel lighter. As one participant explained, *'The main aim of therapy is to share what's in your heart, and your heart feels lighter'*.

Family involvement was considered essential, not optional. Participants emphasized that older relatives often dismissed mental health concerns, which left patients feeling alone or blamed. One participant described becoming exhausted trying to convince an older parent who *'never really agreed there was any such thing'* and stressed the need to *'educate them'*. They recommended bringing family members into sessions, educating them about illness and designating caregivers to supervise medication. This was seen as critical for sustaining adherence and preventing neglect.

Practical concerns shaped how participants imagined interventions working in real life. Many were concerned about scheduling around work, studies and household responsibilities and suggested flexible models such as monthly sessions, short phone calls or video consultations. Simple tools like WhatsApp reminders were viewed as effective ways to reinforce engagement. Finally, participants wanted interventions to extend beyond medical care. They called for support with everyday needs, including dietary guidance, help with employment or small businesses, and spiritual encouragement. For many, health and well-being were inseparable from financial stability, family functioning and faith.

### Theme 4: Social dynamics and support

#### Cultural roles, duties and pressures

Participants' accounts showed how closely their identities were tied to fulfilling culturally expected roles as providers and caretakers. For many men, supporting children, arranging marriages and sustaining household income were central to their sense of self. When mental health faltered, they felt unable to meet these responsibilities, worrying about their children's futures or struggling to keep businesses running. One participant linked his illness to the fear of failing a key paternal duty of arranging his daughters' marriages, saying: *'Now proposals are coming for my daughters, but I cannot speak properly with them'*. Financial stability was described as the foundation of well-being, with income disruptions directly linked to feelings of inadequacy. Women described similar pressures through the unrelenting demands of household work and caregiving. Many pushed themselves to complete daily routines despite exhaustion, determined not to disturb family life. One woman explained: *'Do not feel like doing anything, but I force myself because I'm a housewife, I have kids, a husband'*, emphasizing duty as both motivator and burden. However, these efforts were often met with criticism from spouses or relatives, further straining relationships. As one participant described: *'my husband used to scold me saying you just stay in bed and do not do anything'*, reflecting how symptoms could be interpreted as neglect of expected roles rather than illness.

#### Family and close support dynamics

Participants consistently placed siblings and parents at the center of their support networks, describing them as the first people they could turn to in moments of distress. One participant summarized the value of relational support and emotional expression: *'My brother helped a lot, friends helped too. For me, sharing my feelings was best, it releases tension'*. Mothers were frequently singled out as particularly influential, with some participants crediting maternal care and advice as central to their ability to manage illness. As one participant reflected: *'Most of all, my mother, if I'm okay in my home today, it's because of my mother, she would explain things related to this illness to me'*. Spousal support, however, was less consistent. For some, husbands or wives played an active and supportive role, ensuring treatment was followed and accompanying them to appointments. In other cases, spousal involvement was absent or dismissive, leaving individuals to lean more heavily on siblings or friends. As one participant noted: *'If I tell my wife she'll come, but she's not interested, she says she does not see any need for it'*, highlighting both disinterest and minimization. Among engaged spouses, participants noted that clear explanations from doctors helped partners understand the illness, which, in turn, improved their ability to provide support. One participant emphasized this shift: *'My husband understood because the doctors explained it was an illness, as they guided him, he handled things with me well, and later also informed others that this is an illness'*.

Extended family relationships were often described as more complicated. Some participants encountered conflict, gossip or interference from relatives, which at times deepened household strain. At the same time, some participants described how strictness or firmness from family members encouraged them to take responsibility for their behavior and contributed positively to their recovery. As one participant explained: *'When people dealt with me a bit strictly, I started to change myself'*, suggesting that support could be experienced as both care and corrective pressure within family norms. Overall, the family environment was seen as decisive: a supportive household

could promote recovery, while conflict or neglect risked worsening symptoms.

### Experience of social isolation and anticipated stigma

Participants described how the onset of symptoms often pushed them into isolation from family and friends. Relationships with siblings and extended relatives frequently broke down, sometimes due to direct rejection and at other times because relatives were too preoccupied to offer support. One participant described being openly ostracized: *'All my sisters and brothers cut off contact with me, saying you have gone mad'*. Friendships that once provided companionship became strained, with many withdrawing to avoid conflict or embarrassment. Some noted that their willingness to socialize depended heavily on financial stability and mood, leaving periods of isolation when these conditions were absent. One participant explained: *'If I have money, I socialize happily; if not, I sit in the shop worried'*, showing how distress and financial strain compounded withdrawal.

Fear of stigma played a central role in this withdrawal. Many chose not to disclose their struggles, anticipating ridicule or being labeled as 'mad'. Such derogatory language was reported as deeply hurtful, reinforcing silence and self-concealment. Participants described being repeatedly met with labels like *'you are mental'* or *'you are crazy'*, and one noted that *'the first thing they point out is, "You're mad, your mind is broken, there's no point talking to you"'*. To avoid judgment, participants often presented themselves as outwardly fine, even while struggling privately. In some cases, stigma was embedded in family conflict and gendered interpretations, with symptoms dismissed as manipulation, particularly by extended relatives. One participant described how sisters-in-law framed her distress as *'just making excuses, trying to control my husband'*. At the same time, a few participants felt that awareness of mental health was slowly increasing. They expressed cautious hope that broader acceptance might one day reduce the barriers they faced in maintaining family and community relationships.

### Theme 5: Coping strategies and meaning-making

Across interviews, participants consistently described keeping themselves occupied as a central strategy for managing symptoms of BD. Structured routines, employment and household responsibilities were identified as important ways to regulate mood and distract from intrusive thoughts. For men, formal work and daily labor offered stability and a sense of purpose; for women, domestic tasks such as cooking and cleaning provided both structure and distraction.

Beyond routine responsibilities, individuals also turned to creative outlets and physical activities to support recovery. Painting, craftwork, walking and spending time outdoors were described as restorative and mood-stabilizing. One participant noted: *'for me, physical activities and creative work like painting, art are helpful'*, while another emphasized the value of changing one's environment: *'going out, walking, taking the kids to the park, these things help a lot'*. Some emphasized the value of reconnecting with skills or hobbies learned earlier in life, finding that re-engagement with meaningful activities provided continuity and purpose.

Alongside these activity-based strategies, religious practice formed a central pillar of coping. Structured prayer and Quranic recitation were deliberately used to manage mood, improve sleep and counter intrusive thoughts. One participant described a nightly routine of recitation to restore calm: *'I began listening to Surah Rehman every night to sleep and that brought me peace'*. Certain verses were repeated at night to bring calm, while daily supplications were incorporated into routine as a preemptive shield against relapse. Participants often linked the regularity of these practices to a tangible sense of relief and emotional steadiness.

Alongside behavioral strategies, participants actively sought information about their condition. Many turned to online resources, including websites, YouTube and digital articles, to better understand the causes, symptoms and treatments. While this pursuit of knowledge was seen as empowering, it could also become overwhelming due to the sheer volume of available information. One participant captured this overload: *'there's so much knowledge these days, even months are not enough to read all the articles out there'*. Still, participants linked informed self-management to a stronger sense of control, and some reported that combining what they learned with ongoing therapy helped reduce the severity of depressive episodes.

### Caregivers of persons with bipolar disorder

Thematic analysis of caregiver interviews revealed four central themes reflecting the complex realities of supporting individuals with BD in Pakistan. These themes highlight the intersection of personal sacrifice, cultural pressures and systemic challenges. Illustrative quotes for each theme and subtheme are provided in Supplementary Table 2, in both Roman Urdu and English translation.

### Theme 1: Lived experience of illness and recovery

### Recognition and onset of symptoms

Caregivers described their first encounters with illness as confusing, often beginning with dramatic or puzzling changes in personality and behavior. In many cases, early signs were dismissed as normal stress or part of student or work life. As one caregiver explained: *'At first, it seemed normal, in student life sometimes you get stressed, but then I noticed changes, he became much more aggressive, slept less. That's when it hit me'*, marking the moment everyday stress no longer explained what they were seeing. Families recounted how previously affectionate and respectful individuals became unrecognizable, marked instead by rudeness, aggression and emotional distance. *'He used to be innocent, respectful, loving, all of that disappeared. He became rude, abusive'*, one caregiver said, capturing the grief of losing a familiar relationship. Such abrupt shifts often deepened caregivers' distress, as they struggled to reconcile the person they once knew with the starkly altered version before them.

Caregivers also recalled their own lack of understanding at these early stages. Many admitted that they had little awareness of mental illness and initially turned to friends or family in attempts to interpret what was unfolding: *'I discussed it with my friends' group, we tried to figure out what could be done'*, one caregiver reflected. It was only after repeated crises or worsening symptoms that they came to recognize the need for professional intervention.

Alongside psychological and behavioral changes, families became acutely observant of physical signs, such as changes in facial expression, eye redness or darkening of the face, that signaled impending deterioration. One caregiver described an embodied early warning system: *'I could tell from his eyes, I would inform everyone at home his condition was worsening'*, while another noted, *'Earlier his eyes would get red, and his face would get dark'*. Symptoms were diverse and disorienting. Aggression, rapid irritability and destructive behavior were often the earliest red flags. Caregivers described loss of control and property damage, *'When he got angry, he lost control, broke mirrors'*, and in some cases escalating safety concerns, including *'he would throw whatever was in his hand,*

*sometimes hitting others or himself'*. Others noticed extreme withdrawal, refusal to eat or drink, or a complete halt in communication, which sometimes escalated into conflict at home. Perceptual disturbances, such as paranoia or hearing voices, further compounded the burden of caregiving and heightened uncertainty about what was happening. Caregivers described moments that felt frighteningly unfamiliar: *'After about two years, he started looking at the sky and talking to himself'* and *'he said someone is hitting him or he hears voices, kept thinking someone would do something bad to him'*.

### Cultural and religious interpretations of illness

Caregivers often framed mental illness through culturally grounded models rather than biomedical explanations. Many initially dismissed symptoms as ordinary stress or modern-day problems, attributing changes to lifestyle factors such as sleep deprivation, substance use or past injuries. As one caregiver noted, *'in modern times there are many problems'*, reflecting how distress was initially normalized rather than medicalized. In the absence of medical clarity, supernatural and spiritual beliefs frequently filled the explanatory gap. Illness was described in terms of possession, black magic or the influence of evil forces, reflecting long-standing cultural narratives that made sense of otherwise bewildering behavior.

Marriage emerged as both a perceived cause and a culturally sanctioned remedy. In some cases, marriage was arranged in the hope that it would stabilize the individual. However, these unions often carried their own pressures, such as inadequate dowries, strained relationships or cycles of divorce and remarriage. Family honor and gendered expectations strongly shaped these explanations, with failures in household duties, child loss or marital breakdowns described as both consequences and causes of illness. Families often saw the absence of spousal support as contributing to deterioration; for example, one caregiver said plainly: *'His wife did not support him, she did not go to the doctor'*, reinforcing the belief that relational stability could determine health outcomes. Cultural attitudes toward treatment reflected ambivalence. Medication was sometimes viewed as weakening the mind, while erratic behavior was attributed to personal choice or lack of willpower.

Alongside these cultural frameworks, religious practice played a central role in how caregivers made sense of illness and sustained themselves emotionally. Many turned to prayer, Quranic recitation and remembrance of God as a means of regulating anger, calming distress and finding patience. Faith was described as both a coping strategy and a moral imperative, with caregivers drawing inspiration from religious teachings about endurance and divine reward. As one caregiver explained: *'Patience has great reward, Allah will reward us'*, and another drew strength from prophetic examples: *'I remember how the saints, companions, and prophets endured, so should we'*.

### Theme 2: The multidimensional burden of caregiving

### Disrupted rhythms of daily life

Caregivers consistently described how the illness interfered with the structure of everyday life, requiring frequent adjustments to family routines and household functioning. Participants noted irregular sleep patterns, erratic eating habits and unpredictable emotional states, all of which created challenges in maintaining a stable domestic environment. Sleep disruption was repeatedly described as a central disturbance, *'he used to stay up all night'*, one caregiver said, while another noted the ongoing reversal of day–night rhythm: *'now he stays up all night, but in the morning, he falls asleep'*. The need for near-constant vigilance was a recurrent theme, as families monitored behavior to anticipate crises, remove potential hazards and prevent harm. Night-time wakefulness among patients further disrupted collective routines, often depriving caregivers of rest and undermining household stability. As one caregiver summarized, *'in this illness, not only the patient, but the whole family becomes sick'*.

Medication adherence was considered essential for maintaining balance, though resistance or complaints about side effects often complicated treatment. Caregivers frequently described a clear perceived link between consistent medication use and reduced disruption. *'When he started taking medicine regularly, there was improvement, and these behaviors ended'*. Yet adherence was often fragile and required persistent family effort. Caregivers described ongoing resistance: *'he was stubborn, would not take the medicine, said it made him feel worse'*. Substance use further complicated routines and treatment engagement, with caregivers describing it as a major obstacle. *'The main difficulty is convincing him to quit drugs'*, one caregiver said, and another reported receiving strict medical advice that *'if he smokes after taking medicine, then treatment is not possible'*.

### Practical demands and personal sacrifice

Beyond the day-to-day disruptions, caregivers spoke of the extensive practical adjustments required to provide sustained support. Many tailored their work schedules, withdrew from social engagements or forwent career opportunities to remain available at home. One caregiver described how caregiving directly disrupted employment: *'for a month I did not go to work at all, for the next three or four months, sometimes I went, sometimes I did not'*, showing how crises could destabilize income and routine. Others described constant monitoring alongside work responsibilities, using structured check-ins to maintain safety: *'Before going to the office I would meet him, then every two hours I'd call and after returning home, we'd play his favorite games and talk'*, illustrating how care extended across the entire day. Some caregivers relocated or modified their living situation to reduce conflict and create a calmer environment. One family described leaving their own home to live in a rented house *'for his sake'*, reflecting the extent to which caregiving shaped major household decisions. Financial strain was a frequent concern, with some reporting stalled professional growth or accumulated losses due to repeated absences from work. In some cases, caregivers described needing to borrow money to manage practical demands: *'I kept asking around someone gave 5,000 PKR, someone 1,000 PKR'*, reflecting the informal coping strategies families relied on when support systems were limited. Despite these constraints, caregivers framed these sacrifices as an expected part of family duty, often absorbing the logistical and economic costs of caregiving with minimal external support. At the interpersonal level, many emphasized that caregiving required careful emotional regulation, patience and a deliberate softening of approach. *'Even if you make me angry, I explain with the intention to teach'*, one caregiver said, capturing how personal restraint and relational effort were seen as central to sustaining care over time.

### Emotional containment and health strain

Across narratives, caregivers conveyed the emotional labor embedded in caregiving, particularly the need to suppress distress to maintain family stability. Many described deliberately concealing the severity of the situation from elders or children, framing emotional restraint as both a cultural expectation and a protective gesture. One caregiver explained, *'if I told my parents, they would have become very worried'*, while another noted they carefully managed phone conversations so *'others would not feel it'*, fearing that disclosure would trigger 'tension' and worsen elders' health. In

these narratives, 'tension' functioned as a culturally resonant shorthand for chronic psychological overload with clear physical consequences. In this way, caregiving involved not only managing the patient, but also actively buffering the wider family from emotional contagion and stigma. Caregivers spoke of enduring prolonged stress in silence, postponing self-care and navigating persistent tension, often at the expense of their own well-being. Some described moments when their composure collapsed: *'Sometimes I would start crying myself seeing his condition'*, yet these feelings were often quickly re-contained to avoid destabilizing the household. Gendered expectations were especially visible in how women narrated restraint and endurance. One caregiver described choosing silence to prevent conflict: *'I stay quiet and just tolerate it'*, and another framed endurance as the only viable strategy until children grew older: *'as long as it goes on, let it be; when the kids are grown, then I'll see'*.

Alongside emotional labor, caregivers repeatedly described the embodied toll of long-term strain. Reports of memory lapses, fatigue and chronic health conditions, including hypertension and diabetes, were common, with caregivers explicitly linking these to caregiving stress. *'I forget everything when I'm stressed'*, one participant said, while another connected caregiving to severe metabolic and cardiovascular instability: *'I am a patient of diabetes and blood pressure, because of him my sugar goes to 400, 500, 600, then I stay awake with him all night.'* Others similarly described accumulating illness burden, taking medicines for diabetes, blood pressure and paralysis, along with *'forgetfulness, anger, memory loss'* and *'stiffness in my legs'*.

### Theme 3: Community, social networks and coping

#### Family and community support

Caregivers described support as variable over time and responsive to fluctuations in symptoms and household demands. Within families, roles were informal and continually redistributed according to availability. Younger members provided emotional reassurance and practical help during periods of heightened need. *'My son is just 20 but he kept saying, "'I'm here"'*. Local networks also contributed to monitoring and practical support: *'shopkeepers know him, they called and said he is asking for poison'*, illustrating how community members became part of an informal alert system when risk behaviors emerged.

At the same time, this wider network could generate relational strain. Caregivers worried about the ripple effects of symptoms across extended kinship ties: *'What if my uncle gets upset with me because of his behavior'*, reflecting the pressure to manage not only the patient's conduct but also the family's standing and alliances. Joint-family co-residence functioned simultaneously as a resource and a source of strain. Caregivers described how everyday joint-family dynamics could become triggers, especially around co-sisters and household talk: *'She does not like their negative talk, the way they talk against each other'*, and the caregiver framed avoidance as protective: *'If she is not comfortable, it's good not to force her or else her anxiety will increase'*.

#### Stigma, concealment and social withdrawal

Caregivers commonly limited disclosure of illness to protect family reputation and avoid anticipated judgment. Even when families tried to carry on, they acknowledged the certainty of gossip: *'Yes, behind our backs, people do talk'*. Caregivers described withdrawal as both emotional and embodied: reduced going out, reduced contact and reduced tolerance for social settings. One caregiver observed that patients may avoid interactions because *'when meeting people, they may feel inferior and suffocated'*. Importantly, caregivers emphasized that even education did not guarantee empathy: *'Even educated people sometimes act ignorant and treat us badly'*. This perceived unpredictability of judgment made privacy feel safer than disclosure.

Reputational consequences extended beyond the patient to the entire household. Caregivers described direct damage to family standing and prospects: *'My daughters' engagements ended'*, and another caregiver described how children actively adapted their behavior to avoid association-based shame: *'my daughter worries what others will say about her father, tells him to drop her at a distance from school'*. These accounts show stigma becoming intergenerational, shaping children's identities, school routines and emotional safety.

Yet concealment came at a cost. Families repeatedly described handling illness privately, often explicitly refusing outside help: *'My daughters and I decided whatever needed to be done, we would do it ourselves'*. Strained family relationships amplified isolation. Caregivers noted that some relatives resisted understanding, *'many family members do not want to understand'*, and conflict could become openly hurtful, *'His sisters said many hurtful things'*. Even small affirmations were described as scarce but vital: *'a person becomes disheartened; even a small reassurance that things will improve can help'*, paired with the bleak assessment that *'people no longer have sympathy for others at all'*.

### Theme 4: Health system navigation and access

#### Navigating medical and spiritual systems

Caregivers described navigating plural systems of care, often moving between biomedical and spiritual healing. Families frequently combined prescribed medicines with amulets, prayers or dam (Quranic recitation), reflecting a pragmatic approach of *'whatever works'*. Many explicitly framed recoveries as both: *'half is spiritual, half is medical treatment'*, and even clinicians were said to endorse this: *'Doctor said this illness is cured both spiritually and medically'*.

Medical encounters could bring reassurance: *'meeting the doctor brought a lot of peace'*. Yet caregivers also described confusion and limited guidance: *'we had no guidance, did not know where to go', 'I cannot even name this illness'* and *'no one told us that'*. Spiritual routes were described as mixed: some saw clear benefit: *'he got better with an amulet'* while others reported worsening, *'after dam (spiritual healing) her condition worsened'*. Overall, navigation was trial-and-error across systems, driven by the search for stable, trusted care. Some reported worsening with spiritual healing, while others credited it with clear improvement.

#### Barriers to access and participation

Financial hardship was the most consistent barrier, with families often exhausting savings: *'We spent all our money, after that nothing was left'*; borrowing money or discontinuing care when costs became unsustainable: *'It was very expensive, so we stopped'*. Public services were criticized for overcrowding and rushed consultations: *'in government hospitals, there's so much rush that doctors cannot give proper guidelines'*. Travel distances, sometimes several hours, further restricted access, especially in rural areas: *'you must go to Lahore, 50–70 km, takes 2 hours'*, and transport costs forced trade-offs with children's needs: *'the travel fare is too much, I save that money for the kids' fees'*. Hospital admission was frequently recommended but rarely feasible due to high out-of-pocket costs: *'I'd have to pay for the bed, the room, and I cannot afford it'*.

### Intervention needs and preferences

Caregivers expressed a strong interest in accessible, time-efficient interventions, particularly *via* digital platforms such as WhatsApp, YouTube, Facebook and Instagram. They recommended educational content, reminders and short check-ins as feasible formats. Many also emphasized family inclusion and brief psychoeducation for guardians: *'Guardians should be called in and a meeting held to explain things to them'*. Online sessions were often seen as more convenient: *'I find it better if the session is online; it's easier to attend'*, though many still valued face-to-face interactions. Preferences emphasized brevity (15–30 min), flexibility (once or twice a month) and weekend availability. Some voiced skepticism about the feasibility given societal time constraints, *'people might not have time, in our society, no one gives time'*, but overall, hybrid models integrating digital touchpoints with periodic in-person support were welcomed.

## Discussion

This study explored the lived experiences of individuals with BD and their caregivers in Pakistan, highlighting the interdependence of symptoms, caregiving demands, cultural frameworks and systemic barriers. By centering both perspectives, the findings underscore how BD is not only an individual illness but a family and social experience, shaped by cultural expectations, stigma and strained health systems. These insights align with global research documenting the relational burden of BD (Miklowitz, 2007; Reinares *et al.*, 2016) and add culturally specific dimensions relevant to Pakistan.

Among patients and caregivers, there were some shared realities. Both patients and caregivers frequently drew on cultural and religious explanations to interpret illness. Supernatural forces, black magic and the 'evil eye' were invoked alongside biomedical accounts of stress, trauma and lifestyle factors. These explanatory models resonate with existing research in South Asia, where spiritual attributions remain common and often shape help-seeking pathways (Shafiq, 2020; Umer et al., 2025). Religious practices, such as prayer and Quranic recitation, were described as central coping strategies, echoing findings that spirituality provides meaning and emotional regulation in the face of chronic mental illness (Aggarwal et al., 2023). While these frameworks offered solace, they also delayed recognition of BD as a medical condition, contributing to late or inconsistent treatment. This dual reliance of faith-based coping alongside biomedical care illustrates a pragmatic pluralism that reflects both resilience and vulnerability within Pakistani families.

Caregivers reported integrating their relatives' needs into nearly every aspect of daily life, from supervising routines to restructuring employment and finances. This extensive labor, often invisible, is consistent with evidence that families in low-resource contexts act as the primary safety net for people with severe mental illness (Hu et al., 2025; Tesfaye and Demelash, 2025). Gendered expectations compounded the burden: women were expected to provide silent, enduring care while suppressing distress to protect elders and children. Such emotional containment aligns with findings from South Asian caregiving studies, where silence and self-sacrifice are culturally valorized (Naqvi et al., 2021). The physical health costs described by caregivers, including hypertension and diabetes, reinforce international evidence that caregiver burden has measurable somatic consequences (Chang et al., 2009). These sacrifices were accepted as moral obligations rather than negotiable choices. Caregiving was often framed as a lifelong responsibility, shaped by religious values of patience (*sabr*) and familial duty. This moral positioning underscores why interventions must acknowledge caregivers not only as support resources but also as individuals experiencing cumulative burden and health risks.

Stigma emerged as a pervasive force shaping both patient and caregiver experiences. Fear of gossip, reputational harm and damaged marriage prospects led families to conceal illness from relatives and communities. Similar dynamics have been documented across South Asian populations, where psychiatric illness is often interpreted as a familial failing (Shah et al., 2023). Concealment, while protective against immediate judgment, intensified isolation and restricted access to informal or formal support. Joint-family structures magnified these pressures. While co-residence facilitated shared supervision, it also exposed households to scrutiny and interpersonal conflict. Research has shown that in collectivist contexts, stigma operates not only at the individual level but across the family unit, reinforcing shame and withdrawal (Javed et al., 2021). The accounts here demonstrate how stigma constrained both help-seeking and everyday social participation, contributing to caregiver exhaustion and patient isolation.

Participants' reports of long travel distances, overcrowded public hospitals and financial barriers mirror national evidence of severe treatment gaps (Alvi et al., 2024; Rahman et al., 2024). Families frequently resorted to plural treatment strategies, combining biomedical care with spiritual healing, a pattern observed across low- and middle-income countries where services are limited (Patel et al., 2018). These systemic shortages also shaped medication adherence. Distrust of side effects, inconsistent prescriptions from rotating doctors and the absence of psychoeducation reinforced cycles of relapse. Caregivers bore the brunt of these gaps, coordinating care, financing consultations and enforcing adherence, often without adequate guidance or support.

Compared with high-income Western settings, many challenges identified in our study take distinct forms in the cultural context. For instance, Western caregivers of patients with BD often rely more on professional services and have less involvement from extended family, whereas Pakistani caregivers operate within collectivist, multigenerational households that can both amplify support and intensify stress. Stigma in Western contexts, while present, tends to center on individual identity and may not carry the same burden of family 'honor' or marriageability that we observed in Pakistan. At the same time, several themes we identified – such as the emotional toll on caregivers, cyclical stress from recurring episodes and the critical need for psychoeducation – resonate with reports from other South Asian and Muslim-majority settings (Naqvi et al., 2021; Shah et al., 2023). This suggests that some aspects of the BD family experience are shared across cultures, yet their expression is significantly shaped by local social structures and belief systems. The Pakistani context, with its joint-family norms, religious coping practices and limited mental health infrastructure, may intensify these dynamics in ways that differ from both Western settings and even neighboring countries.

### Implications for intervention

The findings highlight the need for culturally sensitive and resource-efficient family interventions for BD in Pakistan. Psychoeducation delivered through accessible formats, such as WhatsApp content or community workshops, may strengthen illness recognition and treatment adherence, aligning with global task-sharing models (Sangraula et al., 2024; Chau et al., 2025). Integrating spiritual and religious elements into psychoeducation could increase acceptability, given their

central role in coping (Hefti, 2011). Components of family-focused therapy (FFT), including communication skills, problem-solving and structured psychoeducation, map closely onto the challenges identified and may be adaptable to local cultural and family systems.

Given caregivers' central role, interventions should also address caregiver well-being through stress-management support, peer connections and routine health monitoring. Hybrid delivery models combining digital check-ins with periodic in-person sessions may offer a feasible and scalable approach in low-resource settings. Efforts must also target stigma at family and community levels, reframing BD as a manageable condition rather than a source of shame.

### Strengths and limitations

This study is among the first in Pakistan to capture perspectives of both individuals with BD and their caregivers, offering a dyadic understanding of illness and care. The qualitative design enabled exploration of cultural and relational dynamics often missed in quantitative work. Recruitment through a national registry enhanced geographic and socioeconomic diversity, and the use of AI-assisted coding added analytic rigor.

Experiences of living with BD and providing care are likely to vary across several factors, including bipolar subtype (BD-I *vs.* BD-II), illness course and duration, caregiver relationship to the patient, caregiving duration, household structure, socioeconomic circumstances and access to mental health services. As this study was not designed to examine subgroup differences, future work with larger, purposively stratified samples is needed to explore how these factors may shape experiences and needs. Participants were recruited from clinical settings, potentially excluding those relying only on traditional or informal care. Interviews were conducted individually rather than as dyads, which may limit insight into interactional patterns.

Although conducted in Pakistan, these findings may inform services for Pakistani diaspora communities, where similar cultural expectations, stigma and faith-based coping persist. Future research should examine how these dynamics shift across transnational contexts to support more culturally responsive models of care.

### Conclusion

BD in Pakistan is lived not only through patients' symptoms but also through the sacrifices, vigilance and resilience of caregivers. Both groups navigate a complex terrain shaped by cultural explanatory models, entrenched stigma and systemic treatment gaps. These findings emphasize that any effective response must address patients and caregivers together, situating care within the cultural, religious and structural realities of Pakistani society. By centering lived experiences, this study highlights opportunities to design interventions that are locally grounded, scalable and responsive to the intertwined needs of families and individuals living with BD.

**Open peer review.** To view the open peer review materials for this article, please visit http://doi.org/10.1017/gmh.2026.10197.

**Supplementary material.** The supplementary material for this article can be found at http://doi.org/10.1017/gmh.2026.10197.

**Acknowledgements.** We thank the Pakistan Institute of Living and Learning (PILL) for recruitment support and logistical coordination. We are sincerely grateful to all individuals with bipolar disorder and their caregivers who participated and shared their experiences. Their contributions made this work possible.

**Author contribution.** **Madeha Umer** led the conceptualization, methodology development, data curation, formal analysis, project administration and visualization, and drafted and revised the manuscript. **Muqaddas Asif** contributed to investigation, data curation, project administration and manuscript revision. A**meer B. Khoso** supported project administration and manuscript review. **Brett D.M. Jones** contributed to conceptualization and manuscript revision. **Iqra Hassan** contributed to investigation and data curation. **Siqi Xue** contributed to conceptualization and manuscript review. **Sonia Langhani** contributed to investigation, data curation and project administration. **Cindy-Lee Dennis** contributed to conceptualization, methodology, supervision and manuscript revision. Farooq Naeem contributed to conceptualization, methodology, supervision and manuscript revision. **Benoit H. Mulsant** contributed to conceptualization, methodology, supervision and manuscript revision. **Nasim Chaudhry** contributed to resources, investigation and project administration. **Nusrat Husain** contributed to conceptualization, methodology, supervision and manuscript revision. **Muhammad Ishrat Husain** contributed to conceptualization, methodology, supervision and manuscript revision. All authors reviewed and approved the final version of the manuscript.

**Financial support.** The authors report no financial relationships or funding relevant to this manuscript.

**Competing interests.** The authors declare no actual or potential conflicts of interest related to this study.

**Ethics statement.** This qualitative study was approved by the National Bioethics Committee of Pakistan (Ref. No. 4–87/NBC-970/23/457). All participants were adults and provided informed verbal consent prior to participation.

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
