## [Reviewer Report]

The article does a great job at trying to put forward a dyadic understanding for Bipolar Disorder. The work and effort is commendable. I feel it has potential for so much more if the author chooses to review and make revisions.

I feel the study needs to be go through again in order to help fill the gaps and improve the conceptualization of the article. It seems that the interviews were conducted by a one individual and then later analysed by another. If that is the case, I feel it brings up the question of how well the interview notes were shared with the data analyst. In a qualitative research, apart from the interview, field notes play a significant role in highlighting any thoughts, feelings or assertions during and/or after any interview session. These notes help add context to the transcribed data. It would be interesting to know how this gap was filled by the researcher.

Keeping in view the great variation of the sample, I am forced to question the sample size of the study. Bipolar I and II Disorder can vary greatly in experience for both the patient and the caregiver. Relationship with the patient of the caregiver can again vary the experience they would have. Adding no controls to these factors make the sample size limited in terms of true and detailed representation of experience. Apart from these, other variables that I believe would add to immense variation in experience would be socioeconomic status, duration of diagnosis, years of caregiver role, household set-up etc. While these factors have been mentioned but since no form of control was added, the sample seems to be greatly heterogeneous for which a larger sample size should be taken for the results to be more meaningful and truly representative.

The method section can be improved by adding details of how many interviews were conducted with each participant. The Braun and Clarke’s (Braun and Clarke 2006) six-phase reflexive thematic analysis framework should be elaborated and discussed further in data analysis. How was the sample size decided? When and how data collection stopped? Participant characteristics should be mentioned in further detail, probably in a tabular form. This would help the reader get a better idea of the sample division based on numerous demographic factors. it should be clear how many patients had Bipolar I Disorder and how many had Bipolar II Disorder? It is mentioned that “The caregiver sample comprised primarily of brothers (n = 4), followed by mothers (n = 3).” What about the remaining caregiver participants?

The trustworthiness of qualitative research is frequently questioned, perhaps because the concepts of reliability and validity are not addressed in the same way as in a quantitative research. Therefore, the data verification procedures are integral to deal with these issues. I could not find any mention of there procedures. Kindly mention the data verification procedures the researchers must have taken in the study to add to the trustworthiness of their qualitative research.

Lastly, I would suggest to improve the result section. A bit more expression in the writing would help, instead of just plain summary of the interviews which seems to be lacking depth. This addition would make the results more meaningful and interesting for the reader.

---

## [Reviewer Report]

This manuscript is a qualitative study of the experiences of people with bipolar disorder as well as their family caregivers in Pakistan. The study is well-conducted and in a timely manner. The research fills a major knowledge gap in the literature by anticipating the culturally entrenched explanatory propositions, care provision and systemic limitations in a low-resource, collectivist context. The strengths include the rigor of the methodology, the transparency of ethics and the reflexive thematic analysis.

The article is well-written and analytical, with the foundation of the pertinent world and regional literature.

Although the thematic organization is quite self-explanatory and well-written, the paper can use more verbatim quotations of the participants (patients and caregivers) to be included in the Results section. The inclusion of more illustrative data extracts that were presented explicitly in terms of key codes and subthemes would increase transparency and enable the reader to give a more thorough evaluation of the analytic process and base the interpretations on the lived experience of the participants.

---

## [Editor Report]

Dear Ms. Umer,

Your submission to Cambridge Prisms: Global Mental Health refers. Following reports by two independent reviewers, we can now share that the decision of “Major Revision” has been reached. Kindly address each point of critique carefully, and provide a point-by-point response. We are looking forward to receiving your revised manuscript. 

Sincerely,

Prof. André Janse van Rensburg

Cambridge Prisms: Global Mental Health

Manuscript ID: GMH-2025-0420

Manuscript Type: Research Article

Manuscript Title: Lived Experiences of Bipolar Disorder and Family Caregiving in Pakistan

Site URL: https://mc.manuscriptcentral.com/prisms-gmh

---

## [Reviewer Report]

The authors have satisfactorily addressed the comments raised in the previous review. Although the small sample size remains a limitation, the manuscript has improved substantially and is now suitable for publication. I wish the authors the best with their work.

---

## [Editor Report]

Dear Ms. Umer,

Following your revisions in response to peer review critiques and inputs, we have the pleasure to inform you that your paper has been accepted for publication. The editorial office will be in touch with next steps shortly.

Sincerely,

Prof. André Janse van Rensburg

Cambridge Prisms: Global Mental Health

Manuscript ID: GMH-2025-0420.R1

Manuscript Type: Research Article

Manuscript Title: Lived Experiences of Bipolar Disorder and Family Caregiving in Pakistan

Site URL: https://mc.manuscriptcentral.com/prisms-gmh